# Assessing feasibility of establishing antimicrobial stewardship programmes in two provincial-level hospitals in Vietnam: an implementation research study

Vu Thi Lan Huong [ORCID],[1] Ta Thi Dieu Ngan,[2] Huynh Phuong Thao,[3] Le Minh Quang,[4] Tran Thi Thu Hanh,[4] Nguyen Thi Hien,[4] Tran Duc,[4] Vu Hai Vinh [ORCID],[4] Chau Minh Duc,[5] Vo Thi Hoang Dung Em,[5] Phan Van Be Bay,[5] Nguyen Thi Thuy Oanh,[5] Pham Thi Thuy Hang,[5] Nguyen Thi Cam Tu,[1] Truong Anh Quan,[1] Thomas Kesteman,[1] Elizabeth Dodds Ashley,[6] Deverick Anderson,[6] H Rogier van Doorn[1,7]

For numbered affiliations see end of article.

**Correspondence to**
Vu Thi Lan Huong;
huongvtl@oucru.org

## ABSTRACT

**Objectives** To investigate the feasibility of establishing hospital-based antimicrobial stewardship (AMS) programmes comprising action-planning, educational interventions and data feedback in two provincial-level hospitals in Viet Nam.

**Design and setting** This was an implementation research using participatory action process and existing resources from the Duke Antimicrobial Stewardship Outreach Network with local adjustments. A national stakeholder meeting and Strengths-Weaknesses-Opportunities-Threats (SWOT) analysis were conducted to identify gaps and potential interventions.

**Participants** Hospital AMS staff implemented activities throughout the study phases. Routinely collected patient data were analysed to support planning, implementation and evaluation.

**Interventions** Hospitals were considered as a complex adaptive system and leveraged their unique characteristics and interconnections to develop 1-year plans containing core interventions (data use, educational training, prospective audit with feedback (PAF) and evaluations).

**Outcome measures** We assessed feasibility using outputs from stakeholder meeting, SWOT analysis, baseline data, planning process and implementation.

**Results** The stakeholder meeting identified three gaps for AMS at national level: supportive policies, AMS training and core competencies and collaboration. At the hospitals, AMS programmes took 1 year for planning due to lack of hospital-specific procedures and relevant staff competencies. Baseline data (January–December 2019) showed variations in antibiotic consumption: 951 days of therapy (DOT) per 1000 days present in the control and 496 in the intervention wards in hospital 1, and 737 and 714 in hospital 2, respectively. During 1-year implementation, clinical pharmacists audited 1890 antibiotic prescriptions in hospital 1 (June 2020–May 2021) and 1628 in hospital 2 (July 2020–July 2021), and will continue PAF in their daily work.

## Strengths and limitations of this study

► Our study used a participatory and teamwork approach involving hospitals as a complex adaptive system with unique characteristics and interconnections that can be leveraged to develop locally tailored interventions.

► We employed intervention strategies that were conducted using locally available staff and resources, specifically clinical pharmacists and routinely collected data.

► In the process of setting up antimicrobial stewardship programmes, we incorporated perspectives and expertise from both policy makers and hospitals at different levels and from international and local partners.

► Contextualisation and tailoring is a strength, but this also affects the study's external validity, and the results cannot be generalised to all healthcare settings as this implementation study targeted community hospitals with medium levels of resources and staff available.

**Conclusion** Our data confirmed the need to contextualise AMS programmes in low-income and middle-income countries (LMICs) and demonstrated the usefulness of implementation research design in assessing programme feasibility. Developing staff competencies, using local data to stimulate actions and integrating programme activities in routine hospital work are key to success in LMICs.

## INTRODUCTION

Despite fast-growing literature on the effectiveness and potential economic impact of antimicrobial stewardship (AMS) in healthcare facilities around the world,[1–5] limited

data from hospitals in low-income and middle-income countries (LMICs) show numerous challenges in these settings and that implementation must be tailored to the local contexts to have an effect on antibiotic use and antimicrobial resistance (AMR).[6 7] Vietnam, like many other LMICs, is affected by high and increasing levels of AMR. Surveillance data from Vietnam's AMR surveillance network show that the prevalence of antibiotic resistance is among the highest in Asia and the world.[8–10] Hospital-wide antibiotic consumption data from this hospital network also showed critical broad-spectrum and last-resort antibiotics are increasingly used as empiric treatment. In a 12-month point prevalence survey at 15 adult intensive care units within the AMR surveillance network in 2012, carbapenems were the third most commonly used class in all patients and the most common among those with hospital acquired infections.[11]

The Ministry of Health (MoH) in Vietnam developed a National Action Plan on AMR in 2013 for 2013–2020. Surveillance and AMS programmes were two of the six listed priorities of the plans. A nation-wide AMR hospital surveillance network and a reference laboratory have been established.[9 12] A national AMS implementation guideline was issued in 2016[13] and updated in 2020[14] which required each hospital to establish a multidisciplinary AMS team and implement a set of recommended activities. However, many hospitals continue to struggle to implement AMS despite implementation guidance from the Vietnam MoH.[15] In a recent survey of 655 hospitals across the country by the MoH in 2019, 144 (22%) had developed treatment guidelines based on local evidence, 269 (41%) had adopted the preauthorisation policy (antibiotics requiring approval before use for patient treatment), 189 (29%) developed clinical microbiology guidelines and 458 (70%) employed basic infection control policy in their facilities.[15]

In a previous qualitative study about AMS implementation in seven hospitals in the national AMR surveillance network,[12 16] we identified the following barriers and opportunities that hospitals in Vietnam have encountered in setting up and running the programmes: a lack of dedicated resources, ineffective communication between clinicians, pharmacy and microbiology departments, prescribing habits among doctors, including not taking diagnostic samples before starting antibiotic therapy, not adjusting antibiotics based on culture results and hesitation and lack of confidence in de-escalating and switching from intravenous to oral therapy. Antibiotics were mainly prescribed based on clinical presentation and progress, doctors' experience, the requirements for health insurance reimbursements and availability of specific drugs at the pharmacy department (unpublished data). Limited data are available on the process of implementation and impact of AMS programmes in Vietnam; only one report from a large hospital has indicated AMS activities could improve patient outcomes.[17]

In this study, we aim to investigate the feasibility of establishing hospital-based AMS programmes comprising action-planning, educational interventions and data feedback in two provincial-level hospitals in Vietnam. We describe the process of preparation, planning and implementation and identify the factors that are important in setting up such a programme in hospital settings in the country.

## METHODS
### Study design and setting
In this implementation research study, we employed a participatory action process to establish AMS programmes at two provincial-level hospitals in Vietnam. In this process, hospitals were considered a complex adaptive system consisting of evolving inter-related individuals and non-linear interactions, and the study was conducted under the assumption that interventions cannot be applied in different hospitals with predictable results.[18] The process also included assessments at both national and local levels, and adapted from existing resources from Duke Antimicrobial Stewardship Outreach Network (DASON) in Durham, North Carolina, USA and local guidelines. The implementation period of AMS programmes was between 29 July 2020 and 28 July 2021 in one hospital and 1 June 2020–31 May 2021 in the other hospital. Baseline assessments were conducted during 1 year before the implementation period, and evaluation activities were integrated throughout the implementation process.

### Implementation approach
Our planning process began at a national AMS stakeholder meeting organised by the Medical Services Administration of the MoH, WHO and Oxford University Clinical Research Unit (OUCRU), during which we reviewed the current status of AMS implementation in the country. Based on our pre-existing understanding and network with local hospitals, on this meeting, and on subsequent discussions with MoH, we invited two provincial hospitals to join this implementation study: Viet Tiep Hospital in Hai Phong, a large port city with 2 million inhabitants, in the north (2000-bed capacity, 400 000 out-patient and 100 000 in-patient visits per year) and Dong Thap General Hospital in Cao Lanh, Dong Thap in the Mekong Delta in the south (1000-bed capacity, 710 000 out-patient and 65 000 in-patient visits per year). Hereinafter, these two study hospitals will be intentionally referred to as hospital 1 and hospital 2 (in no particular order) to avoid identification of the hospital name in the data presented.

The process of setting up an AMS programme at each hospital went through cycles of planning, implementing and evaluating phases (figure 1). During discussions between the hospitals and the study team, we identified elements of an AMS programme that should be common to both hospitals and elements that could be adapted locally. The results of interventions over time were monitored through repeated evaluations of prescription practices, knowledge and attitudes every 3 months, and interventions could evolve over time to achieve

## AMS implementation cycle at hospital level

## Partnerships in AMS implementation

**Figure 1** Approach to AMS implementation in provincial hospitals and roles of partners. AMS, antimicrobial stewardship

the maximal impact. The AMS team regularly used the collected data and discussed with the study team to identify issues and opportunities for improvement.

Each hospital selected eight clinical wards to participate in the study: four for AMS intervention and four as control wards (the number of wards was determined by the funding availability). The selection of intervention and control wards was based on discussions within the hospital AMS team with two selection criteria: (1) the amount of antibiotic used was greater than average of all wards in the hospital based on pharmacy reported data and (2) the head of ward was willing to participate in the study intervention or control group. The control wards were included to support evaluation of overall AMS outcome indicators (total antibiotic use and for main classes) using trend analysis over 12 months before and after the start of intervention. For practical reasons, there was no strict requirement to identify control wards that were similar to the intervention wards. After eight wards were identified, the decision on the wards to be in the intervention group and control group was left at the discretion of the director board and the AMS team in each hospital. At both hospitals, the directors and the AMS teams decided to select the wards that were more likely to accept the AMS interventions based on their judgement and internal relationships. The allocation of the wards to each study group was as follows:

At hospital 1:

► Intervention group included surgical intensive care unit, traumatology, respiratory/musculoskelectal system, and infectious diseases (ID) ward.
► Control group included internal intensive care unit, surgical nephro-urology, paediatrics and general internal medicine ward.

At hospital 2:
► Intervention group included surgical intensive care unit, general internal medicine, traumatology and high-quality-serviced ward (provide treatment services with better quality based on patient's request).
► Control group included internal intensive care unit, oncology, ID and surgical gastroenterology ward.

The preparation and implementation process started with the training and mentoring component, which was then integrated throughout the study life cycle. Training, technical support and mentoring activities were modelled to international AMS models and developed with experts from DASON, and included sharing experiences and identifying opportunities for local changes and interventions to improve antibiotic use and patient outcomes. DASON is community hospital network that supports the implementation of local AMS programmes and collaborative network participation has shown reductions in antibiotic use in these hospitals.[19 20]

### Training and mentoring

Training and mentoring by local experts on ID, clinical pharmacists and clinical microbiologists to support

AMS implementation were also integrated into the programmes. These experts were from National Hospital for Tropical Diseases, Hanoi (NHTD); Hospital for Tropical Diseases, Ho Chi Minh City (HTD), Vietnam National Children's Hospital, Hanoi and Hanoi University of Pharmacy, Hanoi.

The following training events were organised throughout the planning and implementation phases, with topics discussed and agreed on by all partners involved and timing decided based on practical considerations, including the impact of COVID-19 on routine care in each location:

▶ A study tour to Duke University was organised for AMS staff of two study hospitals and local AMS experts from NHTD and HTD in July–August 2019. Participants attended lectures presented by DASON experts on AMS principles and practical guidance and experiences on specific AMS interventions. Participants also visited four DASON community hospitals to learn about the AMS models implemented at these sites. Participants also discussed about the feasibility and applicability of the interventions to the Vietnamese context.

▶ Visits by DASON experts and OUCRU staff at each study hospital in Vietnam were organised in December 2019 to provide on-site training and technical support in assessing the needs and planning of activities.

▶ Training courses by local experts on antibiotic treatment, clinical pharmacy and clinical microbiology were organised in a class-room style for doctors, pharmacists and microbiologists. These courses were provided at two time points: one initial course in June 2020 to cover the broad concepts and one follow-up course in October 2020 focusing on the specific needs and knowledge gaps of each hospital.

▶ One-week training and shadowing for clinical pharmacists on prospective audit and feedback at HTD was organised in December 2020.

▶ Training course on microbiological specimen sampling was organised at hospital 1 for nurses in April 2021.

▶ Training and on-going technical support were provided by OUCRU for AMS staff on specific skills in planning and implementation of AMS activities.

Local meetings were organised in each hospital with the participation of AMS team members and participating intervention wards to assess the current gaps and needs and develop plans for implementation. Mid-term meetings and project monitoring meetings were also organised by OUCRU to review on the progress and upcoming activities with the AMS team at each hospital.

### Evaluation as part of the implementation cycle

Evaluation was integrated throughout starting with the first round of data collection and analysis before the AMS intervention and every 3 months after programme implementation. The data were used to design and tailor the intervention activities towards the goal in reducing unnecessary antibiotic use in each hospital. Following baseline indicators were evaluated in each hospital (online supplemental table 1):

▶ Antibiotic use in total and from targeted agent groups.

▶ Knowledge and perceptions of doctors.

▶ Proportions of antibiotic prescriptions which had an indication documented, had review/stop date documented, were used as surgical prophylaxis for greater than 24 hours, were not compliant with guidelines currently endorsed at the hospital, and were inappropriate.

▶ Proportion of resistant isolates for priority antibiotic-organism combinations.

▶ Cost of antibiotic treatment and hospitalisation.

▶ In-hospital mortality.

▶ Hospital length of stay.

Data were collected from the existing routine hospital information systems for 12 months baseline period, knowledge-attitude-practice (KAP) surveys of all doctors in eight study wards and relevant AMS staff (pharmacists and microbiologists), and reviews of medical charts in the four intervention wards by the doctors at each of the wards.

The key process indicators of the project were: (1) a functioning AMS team was formed and was able to lead the AMS activities; (2) a 1-year implementation plan for AMS programme was developed and carried out and (3) baseline data and subsequent data from evaluation rounds were used to inform planning and implementation. We recorded implementation data to monitor the intervention activities, specifically prospective audit and feedback (PAF) implemented by the clinical pharmacists. Other data about trainings and meetings (quantity, number of participants, content) and resources required for the activities (costs, time, staff) were also recorded for monitoring the process of programme implementation and support parallel economic evaluation.

### Data analysis and feedback

We reviewed contextual information collected from the national meeting and the baseline needs assessment at the two hospitals and performed an analysis of strengths, weaknesses, opportunities and threats (SWOT) of AMS implementation to support the AMS team in initial planning of the activities at each hospital. We used the following DASON recommendations for the key AMS components to guide our analysis, which include the core elements described by the US CDC[21] and the global recommendations.[22]

#### Leadership

An individual who is qualified through education, training or experience in ID and/or antibiotic stewardship as leader of AMS programme; committee structures for oversight and executive authority for AMS activities; formal policy outlining AMS programme

## Pharmacy

Automated systems (automated dispensing system, barcode medication administration, computerised physician order entry, electronic medical record, electronic medication administration record); pharmacy-based dose optimisation activities; regular antibiotic use evaluations; antibiotic formulary; pharmacy staffing with clinical pharmacist with specialised training and interest in ID

## Microbiology

Laboratory reporting system; cumulative antibiogram; microbiology personnel; testing and incidence of multidrug resistant organisms.

## Infection management

ID clinician presence

We performed descriptive data analysis for each hospital on baseline data and data from the subsequent evaluation rounds to identify issues that required attention and opportunities for intervention in the AMS programme. Results were summarised and presented to provide feedback to the hospital management board, AMS team and intervention wards through reports and project progress meetings and to use as evidence for identifying further actions. We characterised antibiotic use by subgroups, types of clinical wards and international classification of diseases (ICD-10) code groups. Hospital-wide antibiotic susceptibility testing results entered into WHONET were extracted and analysed following the same analysis protocols performed in previous studies for AMR surveillance data from VINARES hospital network.[9 10] We reported microbiology data following the recommendations of the MICRO checklist[23] as shown in online supplemental table 2.

Resistance results were expressed as proportions of resistant isolates out of the number of tested isolates. We classified multidrug resistance (MDR) and carbapenem resistance based on the criteria proposed by Magiorakos et al[24] as follows:

► For *Acinetobacter baumannii* and *Acinetobacter spp.*: MDR is defined as resistant to at least three of the following: cephalosporin (ceftriaxone or cefepime), aminoglycosides (amikacin, gentamicin or tobramycin), ciprofloxacin and carbapenem (imipenem or meropenem); carbapenem resistance is defined as resistant to imipenem or meropenem.

► For *Escherichia coli, Klebsiella pneumoniae* and *Klebsiella spp.*: MDR is defined as resistant to at least three of the following: carbapenem (ertapenem, imipenem or meropenem), cephalosporin (ceftriaxone or cefepime), aminoglycosides (amikacin, gentamicin or tobramycin) and ciprofloxacin; carbapenem resistance is defined as resistant to ertapenem, imipenem or meropenem.

► For *Pseudomonas aeruginosa*: MDR is defined as resistant to at least three of the following four agents: imipenem, ceftazidime, ciprofloxacin and tobramycin; carbapenem resistance is resistant to imipenem or meropenem.

## Patient and public involvement

Patient and public involvement was not included in this study as this is a new aspect in the local hospital settings and requires additional research to explore its feasibility and the most appropriate approaches. Initial efforts are being undertaken in parallel to this research for involving patients and the public in AMS in the local hospitals.

Here we present the results of the initial steps in setting up the programme and describe the baseline summary indicators from the routine data of the two hospitals. This paper was reported following the StaRI checklist 'Standards for Reporting Implementation Studies: the StaRI checklist for completion'.[25]

## RESULTS

### Baseline analysis from the national AMS stakeholder meeting

The meeting was organised in March 2019 to review the current state of AMS implementation nation-wide with participation of a wide range of stakeholders representative for both public and private sectors, and governmental and international partners working or interested in AMS implementation (online supplemental table 3). The meeting opened a platform for shared discussions on different aspects of implementation and identified the three major areas for actions at the national level: strengthening a supportive policy environment for AMS, developing AMS training curriculum and core AMS related competencies, and establishing coordination and collaboration in AMS within Vietnam and the region. Further stakeholder meetings were recommended for continued sharing of experience and promote networking and collaboration in AMS locally and regionally, with one such meeting subsequently organised in December 2019 by OUCRU.

### Baseline analysis at two hospitals and implementation plan

Data on existing processes and structures were reviewed to provide input for initial planning of AMS intervention. In table 1, we summarised the results of a SWOT analysis conducted during the preparation phase at both hospitals in 2020 of the current state of key components, the needs identified for each component and the corresponding AMS interventions to be implemented. There was a large gap between the baseline situation in each hospital and the recommendations by DASON in all components. Leadership commitment was identified as a prerequisite for site selection, however the level of participation was different between the two hospitals. At one hospital, a vice director directly leads the AMS programme together with the General Planning Department, which drives the implementation activities and encourages the clinical wards to actively be engaged in the programme. The process was slower at the other hospital where the programme was

**Table 1** SWOT analysis of current status of key components, assessment of needs and interventions implemented at two provincial hospitals

| Component | SWOT analysis (strength–weakness–opportunity–threat) | Assessment of needs | Interventions implemented |
|---|---|---|---|
| **Hospital 1** | | | |
| Leadership | S: leadership commitment; vice-director had an active role in AMS committee<br>W: no AMS experience, large committee with no dedicated AMS coordinator for day-to-day activities<br>O: guidelines from MoH and experience from other higher level hospitals; leadership intention to provide AMS outreach support to district hospitals in long term<br>T: disease outbreaks (eg, COVID-19) shifting management priorities | Knowledge and skills in planning and implementing AMS<br>Identify clear roles in planning and implementing AMS activities<br>Identify targets for interventions | Establishing AMS action team for day-to-day activities with active coordination from Planning Department<br>Regular minuted team meetings<br>Training for AMS committee by international and local experts<br>Collecting and reviewing data to inform interventions |
| Pharmacy | S: drug inventories and bidding<br>W: no clinical experience; only tracking drug purchasing but not actual drug use data; lack of analytical skills<br>O: electronic data on patient-level drug administration available from clinical database; government policy and guidelines (clinical pharmacy, AMS guidelines)<br>T: high staff turnover, lack of clinical pharmacists, bidding process and insurance limits (can affect drug choices and prescribing behaviours) | Identify staff for clinical pharmacist roles<br>Build clinical pharmacy capacity and data analysis skills | Training for clinical pharmacists in antibiotic treatment, microbiology, pharmacology and review of antibiotic prescriptions (class-room, hands-on)<br>Implemented prospective audit and feedback as a routine activity of clinical pharmacy programme<br>Participation of clinical pharmacists in clinical ward monthly meeting for which they had to analyse audit data and present to the doctors<br>Support from DASON on using web-based data visualisation and benchmarking tool* |
| Microbiology | S: in-house microbiology laboratory; technical support by OUCRU; use WHONET database<br>W: contamination issues, low test utilisation; lack of clinical interactions; no clinical microbiologist; performing manual culture; limited quality control<br>O: close connection with OUCRU laboratory; a new laboratory under construction<br>T: lack of compliance in sample collection from clinical wards; weak voice in planning and ordering reagents and tests; insurance limits (can affect culture ordering behaviour) | Improve quantity and quality of routine clinical specimens, and quality of microbiology testing<br>Active planning from microbiology laboratory<br>Active communication from laboratory to clinical wards on patient's specimens and test results<br>More regular tailored updates on local AMR data to support doctors in empiric treatment | Training on clinical microbiology for laboratory staff by locally recognised experts<br>Review/update Standard Operating Procedures (SOPs) with support from OUCRU laboratory<br>Nurse training on collecting clinical specimens<br>Support from OUCRU laboratory to review the current microbiology practices and planning for improvement<br>Analysis of WHONET data twice yearly and creating readable summary of AMR patterns to doctors |
| Infection management | S: hospital-wide consultations with ID doctor available; head of ID department is in the AMS committee<br>W: limited access to training<br>O: established connection with Hospital for Tropical Diseases in Ho Chi Minh City<br>T: disease outbreaks (eg, COVID-19) | Update knowledge and skills in management of specific and locally relevant infectious diseases, clinical pharmacology and antibiotic treatment guidelines | Training on infectious diseases management and antibiotic treatment for main clinical syndromes, principals of clinical pharmacology and microbiology and surgical prophylaxis |
| **Hospital 2** | | | |

Continued

**Table 1** Continued

| Component | SWOT analysis (strength–weakness–opportunity–threat) | Assessment of needs | Interventions implemented |
|---|---|---|---|
| Leadership | S: leadership commitment<br>W: no AMS experience, large committee with no dedicated AMS coordinator for day-to-day activities<br>O: MoH guidelines and experience from other higher-level hospitals; leadership dedication to high-quality hospital performance<br>T: disease outbreaks (eg, COVID-19) shifting management priorities; hospital's focus on surgical services and cancer treatment | Knowledge and skills in planning and implementing AMS<br>Identify clear roles in planning and implementing AMS activities<br>Raise the awareness and stimulate interests of surgical doctors in AMS<br>Identify targets for interventions | Establishing AMS action team for day-to-day activities with active coordination from Planning Department and International Relations Department<br>Select four surgical wards to participate in the study (two intervention and two control wards)<br>Training for AMS team by international and local experts<br>Collecting and reviewing data to inform interventions |
| Pharmacy | S: drug inventories and bidding; young pharmacists available; hospital has interest in improving clinical pharmacy programme<br>W: limited clinical experience; only tracking drug purchasing but not actual drug use data; lack of analytical skills<br>O: electronic data on patient-level drug administration available from clinical database; government policy and guidelines (clinical pharmacy, AMS guidelines); close connection with the Hanoi University of Pharmacy for clinical pharmacy training<br>T: non-compliance from surgical departments; bidding process and insurance limits (can affect drug choices and prescribing behaviours) | Identify staff for clinical pharmacist roles<br>Build clinical pharmacy capacity and data analysis skills | Training for clinical pharmacists in antibiotic treatment, microbiology, pharmacology and review of antibiotic prescriptions (class-room, hands-on)<br>Implemented prospective audit and feedback as a routine activity of clinical pharmacy programme<br>Participation of clinical pharmacists in clinical ward monthly meeting for which they had to analyse audit data and present to the doctors<br>Join doctors in their clinical ward rounds to get more clinical experience and interactions with doctors<br>Support from DASON on using web-based data visualisation and benchmarking tool* |
| Microbiology | S: in-house good-quality microbiology laboratory; technical support by OUCRU and US CDC; use WHONET database<br>W: limited clinical interactions<br>O: close connection with OUCRU laboratory; have access to advanced techniques and skills<br>T: lack of compliance in sample collection from some clinical wards | Increase communication with doctors on patient's specimens and test results<br>Increase clinical experience for clinical microbiologists<br>More regular tailored updates on AMR data to support doctors in empiric treatment | Experience sharing and discussions for lab staff with locally recognised experts<br>Analysis of WHONET data twice yearly and creating readable summary of AMR patterns to doctors<br>Join doctors in their clinical ward rounds to get more experience and interactions with doctors |
| Infection management | S: hospital-wide consultations with ID doctor and specific specialists available; strong experience from ICU doctors in difficult cases<br>W: no presence of ID doctor in AMS committee<br>O: established connection with national-level hospitals<br>T: disease outbreaks (eg, COVID-19); dominance of surgical and ICU doctors in determining antibiotic choices for patients | Update knowledge and skills in management of specific and locally relevant infectious diseases, clinical pharmacology and antibiotic treatment guidelines<br>Increase the interests and awareness of surgical doctors in AMS | Training on infectious diseases management and antibiotic treatment for main clinical syndromes, principals of clinical pharmacology and microbiology and surgical prophylaxis for young doctors<br>Set the target for reducing use of two or more antibiotics for surgical prophylaxis in traumatology department |

*Delayed due to discussions on data sending outside of Vietnam and where the tool could be located.

AMR, antimicrobial resistance; AMS, antimicrobial stewardship; DASON, Duke Antimicrobial Stewardship Outreach Network; ICU, intensive care units; ID, infectious diseases; MoH, Ministry of Health; OUCRU, Oxford University Clinical Research Unit.

managed directly by the General Planning Department and International Collaborations Department.

Both hospitals provide care and treatment to all types of clinical conditions and diseases in the catchment areas. Each hospital has a department of ID, providing care and treatment to patients who are admitted to this department, and provide consultation in managing patients with signs of infection in other clinical wards if required. Each hospital also has a department of pharmacy and a microbiology laboratory. In both hospitals, pharmacists were mostly involved in drug inventory and bidding, and some support to doctors in dose adjustment in special cases, and microbiologists' main roles were to inform doctors of test results. Engagement of microbiology, pharmacy and ID departments in antibiotic treatment was generally passive, mostly through hospital-wide consultation meetings on an ad-hoc basis when there were difficult cases. Local AMR surveillance data were not readily available for use to inform treatment guidelines or updating drug formularies.

From the gaps and needs identified, specific strategies were developed within the first-year plan for interventions targeting each of the key components (table 1). These strategies included:

1. Establishing leadership commitment and forming an AMS coordinating team that can connect all relevant wards and units within each hospital. At both hospitals, the planning department plays a key and coordinating role with a linkage to the specific clinical and functional departments and reporting responsibility to the hospital management board;.
2. Using data to inform planning and evaluating AMS activities and stimulate the actions among leaders and staff through regular AMS meetings organised at each hospital. Data were from the KAP surveys, review of patient medical charts and routine data on antimicrobial treatment, clinical diseases, antimicrobial susceptibility data and denominator data (days present at each study ward);
3. Training/mentoring activities to update the knowledge and skill gaps for doctors in the intervention wards based on the findings from quarterly KAP surveys and retrospective reviews of patients charts for antibiotic prescribing practices. These trainings were delivered by national experts on ID, clinical microbiology and clinical pharmacy; technical training for AMS staff on specific skills including setting up an AMS programme, using data for planning and monitoring, PAF, communication skills and developing guidelines provided by DASON experts and local experts;
4. Implementing PAF at the intervention wards by clinical pharmacists with support from the AMS doctor and team members. On-site training and shadowing experienced clinical pharmacists in HTD during their PAF activities were organised to develop the capacity for these new teams. By the end of the implementation period, 1890 antibiotic prescriptions were audited in hospital 1 (time period: June 2020–May 2021; 1.5 full-time

clinical pharmacists) and 1628 in hospital 2 (July 2020–July 2021; four full-time clinical pharmacists). Recommendations were made in 82/1890 (4.3%) of the audited cases in hospital 1 and 128/1628 (7.9%) in hospital 2. The hospitals planned to continue PAF in the daily work of the clinical pharmacists.

Some strategies differed by site. At hospital 1, due to limited microbiology data available for developing treatment guidelines the hospital decided to focus first on increasing the quantity and quality of specimen cultures through updating the current SOPs and organising trainings for technicians and nurses at each clinical ward. At hospital two where there are more surgical wards, the hospital AMS team put more focus on surgical prophylaxis with an initial aim to reduce the number of two-drug combination orders.

### Baseline summary indicators before intervention
#### Antibiotic use data

Total number of patients admitted at the intervention and control wards at the two hospitals in 2019 and the proportions with antibiotic treatment during the whole hospital admission are summarised in table 2 following the ICD-10 codes for the main diagnosis recorded for each patient. The spectrum of diseases was by definition heterogeneous among the patients admitted at the intervention and control wards but also in both absolute number and the proportions being prescribed antibiotics. The diagnosis groups with highest amount of antibiotic use in DOT in 2019 for both intervention and control group in two hospitals also differed. The top three for each group were (in brackets, the average number of DOT per 1000 days present):

▶ Control group in hospital 1: genitourinary diseases (1057), respiratory diseases (1028) and certain infectious/parasitic diseases (927).
▶ Intervention group in hospital 1: respiratory diseases (897), certain infectious/parasitic diseases (863) and injury, poisoning and certain other consequences of external causes (729).
▶ Control group in hospital 2: digestive diseases (800), certain infectious/parasitic diseases (797) and neoplasms (553).
▶ Intervention group in hospital 2: respiratory diseases (888), injury, poisoning and certain other consequences of external causes (736) and digestive (707).

Figure 2 displays the monthly distribution of antibiotic use in patients in the intervention and control wards. The amount of antibiotic consumption varied significantly by month in 2019 in the study wards for both hospitals: 951 DOT per 1000 days present in the control compared with 496 in the intervention group in hospital 1, and 737 in the control compared with 714 in the intervention group in hospital 2. Hospital 1 used more third-generation cephalosporins than hospital 2, while hospital 2 used more penicillin combinations than hospital 1. In addition, there were more variations across the months in the

**Table 2** Number of patients and proportions of antibiotic use by diagnosis groups at intervention and control wards at two hospitals between January and December in 2019 before the intervention period

| Diagnosis | Hospital 1 | | | | Hospital 2 | | | |
| --- | --- | --- | --- | --- | --- | --- | --- | --- |
| | Intervention wards* | | Control wards† | | Intervention wards‡ | | Control wards§ | |
| | N | % | N | % | N | % | N | % |
| **ICD diagnosis group** | | | | | | | | |
| Certain conditions originating in the perinatal period | 35 | 45.7 | 1292 | 81.4 | 10 | 30.0 | 3 | 66.7 |
| Certain infectious and parasitic diseases | 4756 | 47.3 | 3712 | 79.3 | 308 | 77.3 | 2131 | 59.6 |
| Congenital malformations, deformations and chromosomal abnormalities | | | | | 19 | 47.4 | 8 | 75.0 |
| Diseases of the blood and blood-forming organs and certain disorders involving the immune mechanism | 20 | 60.0 | 405 | 23.2 | 41 | 22.0 | 19 | 47.4 |
| Diseases of the circulatory system | 146 | 79.5 | 1828 | 36.8 | 818 | 35.5 | 223 | 84.8 |
| Diseases of the digestive system | 2568 | 87.3 | 2673 | 25.0 | 676 | 71.3 | 2451 | 83.6 |
| Diseases of the ear and mastoid process | 38 | 81.6 | 26 | 26.9 | 510 | 12.4 | 3 | 33.3 |
| Diseases of the eye and adnexa | 2 | 100.0 | 5 | 100.0 | 7 | 42.9 | 1 | 0.0 |
| Diseases of the genitourinary system | 852 | 63.0 | 1790 | 71.5 | 253 | 93.3 | 82 | 86.6 |
| Diseases of the musculoskeletal system and connective tissue | 473 | 35.3 | 213 | 26.8 | 83 | 25.3 | 17 | 58.8 |
| Diseases of the nervous system | 136 | 72.8 | 115 | 49.6 | 136 | 25.0 | 95 | 86.3 |
| Diseases of the respiratory system | 2039 | 82.9 | 7366 | 95.2 | 813 | 93.0 | 856 | 93.3 |
| Diseases of the skin and subcutaneous tissue | 195 | 87.7 | 433 | 68.4 | 35 | 65.7 | 20 | 90.0 |
| Endocrine, nutritional and metabolic diseases | 203 | 74.9 | 322 | 25.5 | 140 | 30.0 | 39 | 69.2 |
| External causes of morbidity and mortality | 13 | 69.2 | 302 | 35.4 | 5 | 20.0 | 41 | 70.7 |
| Factors influencing health status and contact with health services | 17 | 100.0 | 3 | 66.7 | 6 | 16.7 | 3 | 66.7 |
| Injury, poisoning and certain other consequences of external causes | 3504 | 79.9 | 202 | 40.1 | 327 | 83.5 | 120 | 70.8 |
| Mental, behavioural and neurodevelopmental disorders | 3 | 100.0 | 17 | 52.9 | 35 | 8.6 | 8 | 37.5 |
| Neoplasms | 912 | 79.3 | 409 | 62.6 | 252 | 61.1 | 9186 | 21.8 |
| Pregnancy, childbirth and the puerperium | 3905 | 79.3 | 4 | 100.0 | 7 | 71.4 | 3 | 66.7 |
| Symptoms, signs and abnormal clinical and laboratory findings, not elsewhere classified | 224 | 79.3 | 716 | 76.8 | 178 | 78.1 | 673 | 92.3 |
| **Patients diagnosed with viral causes** | | | | | | | | |
| Dengue | 1503 | 9.2 | 187 | 28.9 | 28 | 21.4 | 408 | 21.6 |
| Influenza | 1310 | 92.1 | 4497 | 87.1 | 574 | 93.0 | 673 | 97.3 |
| Other viral diagnoses | 3213 | 27.1 | 1769 | 63.4 | 169 | 68.6 | 1316 | 40.7 |

*Includes surgical intensive care unit, traumatology, respiratory/musculoskelectal system and infectious diseases ward.
†Includes internal intensive care unit, surgical nephro-urology, pediatrics and general internal medicine.
‡Includes surgical intensive care unit, general internal medicine, traumatology and high-quality-serviced ward (provide treatment services with better quality based on patient's request).
§Includes internal intensive care unit, oncology, infectious diseases and surgical gastroenterology ward.
%, proportion of patients with antibiotic treatment; ICD, international classification of diseases; n, total number of patients.

amount of use for each antibiotic subgroup in hospital 2 than hospital 1.

## Microbiology data

There were 2430 isolates reported from culture of 10 784 specimens (all types of specimens, 22.5% positive rate) in hospital 1, with 289 isolates from 3583 blood specimens (8.1% positivity rate) in 2019. Hospital 2 reported 5463 isolates recovered from 23 731 specimens (all types of specimens, 23.0% positive rate) and 1642 isolates

recovered from 14 539 blood specimens (11.3% positive rate) in 2019.

Among the study wards, there were 1392 isolates (5.5 isolates per 1000 days present) reported for eight study wards in hospital 1 (198 blood isolates, 14.2%) in 2019. Slightly more isolates were from male patients (56.6%) and from older age range (median age 63; IQR 48–74) in the eight wards of this hospital. Hospital 2 reported 2433 isolates (13.4 isolates per 1000 days present) for eight study wards in 2019, of which 802 (33.0%) were

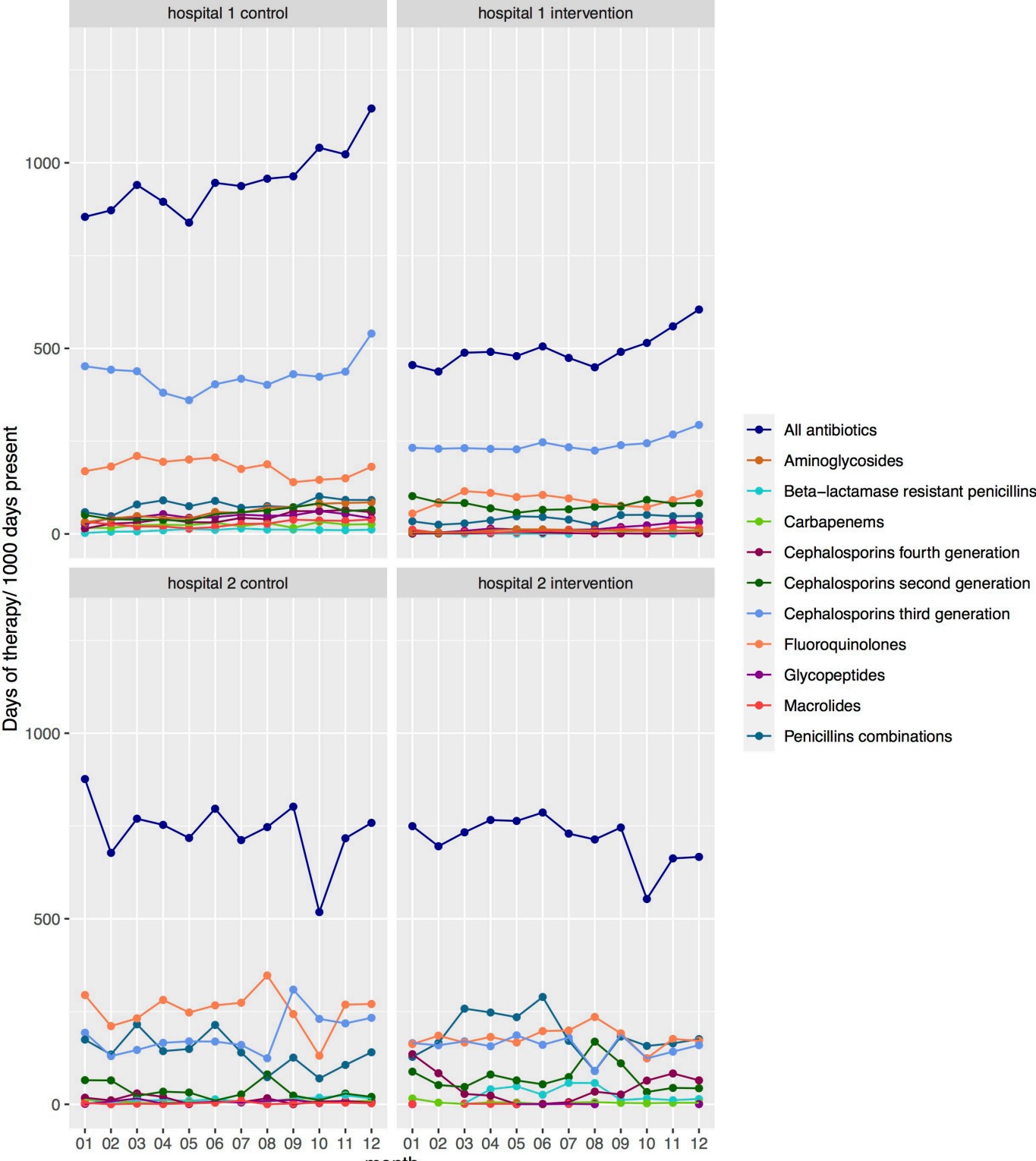

**Figure 2** Baseline monthly distribution of antibiotic use in days of therapy in 2019 calculated for all patients with antibiotic treatment administered during their admission at the study wards in each hospital per 1000 days present at these wards stratified by study group and antibiotic subgroups.

blood isolates. These isolates were also mainly from male patients (64.3%) and older age range (median age 64; IQR 52–76).

*E. coli* was the most commonly isolated bacterium in the study wards in both hospitals (figure 3), and harboured

8.3% (n=540) carbapenem resistance and 34.1% (n=539) MDR (resistant to at least three of the following: carbapenem, cephalosporin, aminoglycosides and ciprofloxacin) in hospital 1, and 7.4% (n=297) and 20.6% (n=267), respectively, in hospital 2 (online supplemental table 4).

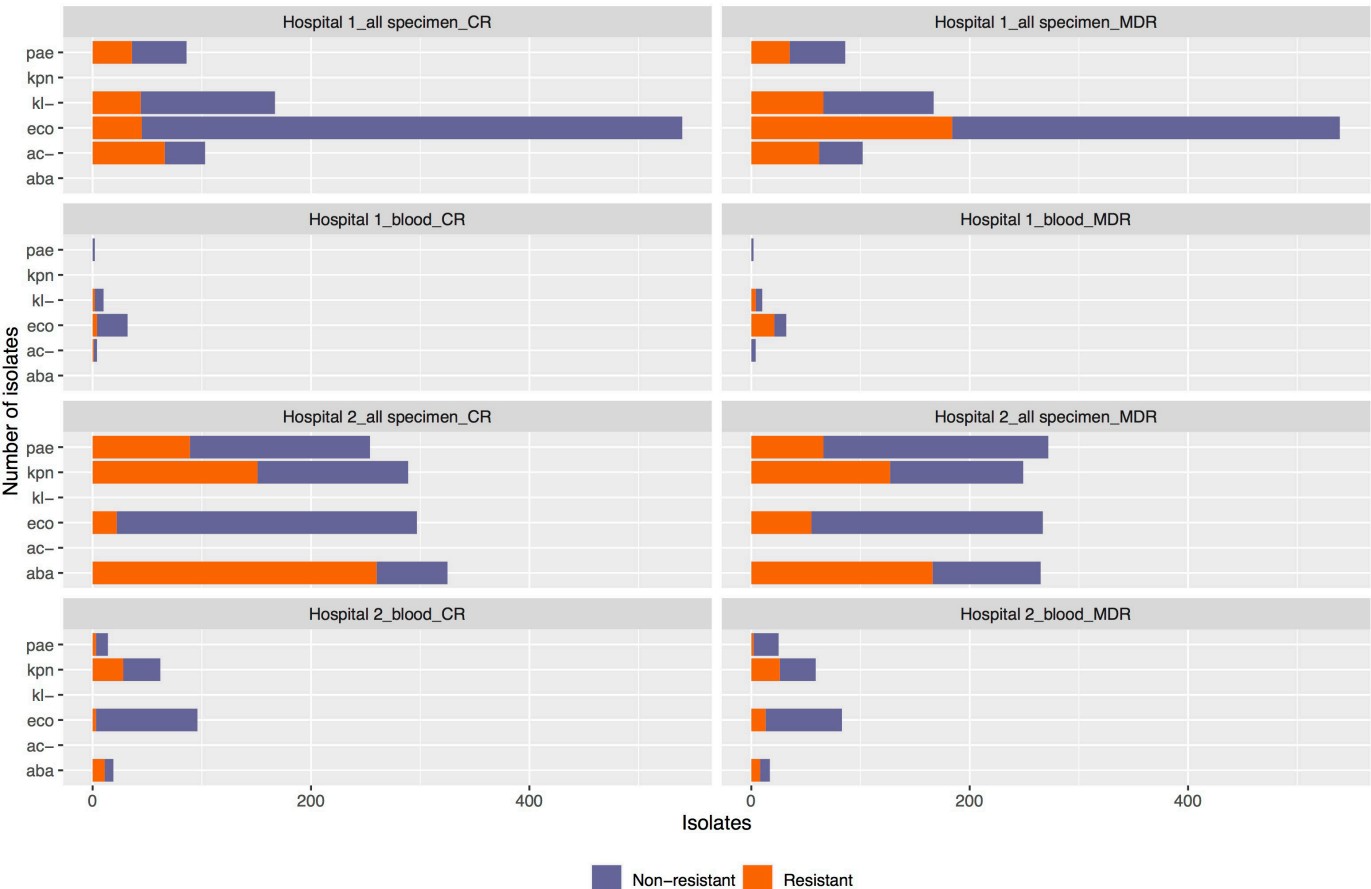

**Figure 3** Carbapenem resistance (CR) and multidrug resistance (MDR) proportions among key bacteria isolated in study wards at two hospitals in 2019. Bacteria include: *Acinetobacter baumannii* (aba) or *Acinetobacter spp.* (ac-), *Escherichia coli* (eco), *Klebsiella pneumoniae* (kpn) or *Klebsiella spp.* (kl-) and *Pseudomonas aeruginosa* (pae).

*K. pneumoniae* in hospital 2 showed a high level of resistance to carbapenem (in all specimens 52.2% n=289 and in blood 45.2% n=62) and MDR (51.0% n=249% and 44.1% n=59, respectively). For hospital 1, *Klebsiella spp.* showed lower carbapenem resistance (26.3% n=167) and MDR (39.5% n=167) in all specimens.

High levels of carbapenem resistance in all specimens (80.0% n=325) and in blood (57.9% n=19) and MDR (resistant to at least three of the following: cephalosporin, aminoglycosides, ciprofloxacin and carbapenem) in all specimens (62.6% n=265) and in blood (47.1% n=17) were reported for *A. baumannii* in hospital 2 and *Acinetobacter spp.* in hospital 1 (in all specimens 64.1% n=103 carbapenem resistance and 60.8% n=102 MDR).

For *P. aeruginosa* in all specimens, the proportions were 25.3% n=253, 35.0% n=254% and 24.3% n=272 for ceftazidime resistance, carbapenem resistance and MDR (resistant to at least three of the following four agents: imipenem, ceftazidime, ciprofloxacin and tobramycin), respectively, in hospital 2, lower than those in hospital 1 (41.2% n=85, 41.9% n=86% and 40.7% n=86, respectively).

Proportions of methicillin-resistant *Staphylococcus aureus* were 79.0% (n=210) in hospital 1 and 76.7% (n=103) in hospital 2 based on the cefoxitin test.

## DISCUSSION

We used an implementation research approach to evaluate the feasibility and understand the process of AMS implementation in two pilot hospitals in Vietnam. First, this study showed that building a functioning multidisciplinary AMS team is an important factor for the success of the programme. In Vietnam, AMS committees have been formed in 334 out of 655 hospitals (51%) in a recent MoH report in 2019; however, the majority of these committees had irregular activity (72%).[15] In our study, AMS teams were led by hospital director/vice-director and the planning department, which has the advising role to the director-board. These combinations were routed from the traditional set-up of Vietnamese hospitals and appeared to work effectively in most hospital-wide programmes. SWOT analysis, a recommended tool in the WHO Resources for AMS programmes in healthcare facilities in LMICs,[6] was used in our needs assessment and has proved to effectively help our hospital AMS teams identify the needs and focus priority interventions.

Second, the partnership approach combining local and international partners in our study demonstrated an effective synergy of experience, expertise and resources to ensure the appropriate selection of strategies and methods in setting up AMS programmes in LMICs. This

approach has been advocated in a recent call for international collaborations in AMS implementation in LMICs, which can be achieved through the dissemination of high-quality research and educational resources recognising the needs to contextualise to the local conditions.[26] Our study provided further evidence to support this call for greater international collaboration and partnerships to facilitate shared expertise and knowledge for AMS.

Across a number of AMS programmes worldwide, pharmacists are often included to manage the prescription-based strategies including preauthorisation formularies and postprescription audits and feedback (PAF). This was reflected in the national guideline on AMS implementation originally issued in 2016 and updated in 2020.[13 14] Sustained reductions have been achieved in antibiotic use by AMS teams across different settings including the example of non-specialist pharmacist-led programme in South Africa[27] and clinician-led and clinical pharmacist-driven AMS programme in India.[28] In our study, despite having no experience in clinical interactions, our junior pharmacists have been building their PAF capacity and gained promising results in the number of audits with feedback to the doctors. Although the results were highly variable between hospitals, this shows the feasibility and potential of involving pharmacists in the prescription-based strategies for AMS programmes in LMICs. A recent prospective multicentre study in community hospitals in the USA has also shown the effective role of pharmacists in implementing the preauthorisation and postprescription audit and feedback in improving antibiotic use in resource-limited settings.[29]

AMS programmes in LMICs generally encounter challenges of limited financial and physical resources available for implementation including substandard lab practices, lack of well-trained staff, insufficient opportunities for continuing professional development and difficulties in deploying prescription-based stewardship strategies.[26 27] Despite numerous challenges and limited, poor-quality evidence available,[30] effective AMS programmes can be feasible in these settings although contextualisation is essential in this implementation.[7] The need for contextualisation in LMICs was also highlighted in several qualitative and mixed-methods AMS studies.[31–33] In our programme, AMS implementation relied on external financial support with the funded intervention period planned for 1 year. Considerations were made regarding choosing the most effective strategies that can support ongoing AMS team operations and continued AMS activities after the funding period ends.

Among these considerations, capacity building and training for the AMS team in programme planning and evaluation and hospital pharmacists to conduct PAF activities were identified as the key components. As recommended in a recent review of AMS programme designs, practically integrating the planned changes into the normal routine clinical practices is the best way to achieve sustainability apart from securing long-term resources.[34] Making PAF activities routine can contribute to the sustainability of the AMS programme. In addition, developing methods and staff capacity to utilise the existing routinely collected data (clinical, pharmacy and microbiology) was also a priority in our programme. Nevertheless, at both study hospitals, the outstanding issue remained in the use of local evidence to inform practices in antibiotic treatment and management, and a lack of information technology and epidemiological capacity partly contributes to this issue. This issue will need to be addressed in the next longer-term AMS programme plan at both hospitals.

AMS programmes have been traditionally 'pharmacy-centric' with minimal or no input from microbiology; participation of clinical microbiologists in AMS teams varied widely around the globe (20%–90% across the surveys in Europe, Australia, USA and Canada).[35] Microbiology capacity and surveillance and control of MDR bacteria is a common challenge in AMS implementation for hospitals in LIMCs,[34 36] and this is particularly the case for provincial and community level hospitals in Vietnam. From the MoH survey in 2019, 90% of hospitals had a pharmacist, while only 39% included a microbiologist in their AMS team. The current status of microbiology in LMICs would not allow for a desirable level of support to AMS programmes.[26] However, improvements can be made even within resource-constrained conditions. Our study showed that simply compiling the current data and presenting the summary data in the AMS meetings has stimulated the discussions on the immediate and long-term actions to improve the problem. The participants realised the need to have good microbiology data as a foundation for more effective and targeted interventions, and the first simple step was to review the procedures, organise trainings and encourage doctors to perform specimen cultures prior to initiating antibiotic treatment. Such changes have been made feasible by an active AMS team with motivated team leaders which is considered as a prerequisite for the success of AMR containment interventions in LMIC settings.[37]

The strengths of our study include the participatory approach in AMS implementation where hospitals were considered as a complex adaptive system with unique characteristics and interconnections between players that could be leveraged to develop locally tailored interventions for maximal results.[18] These included identifying strategies that could be sustained relying on local resources as routine practices including PAF and data use. Second, the study has employed interdisciplinary and team activities in both implementation and evaluation of AMS programme, and emphasised the need to improve the capacity of all stakeholders involved in each hospital setting. Third, our implementation approach was designed to include perspectives and expertise from both policy makers to hospitals at different levels and from international and local partners. Our data from the implementation process were consistent with and consolidating the outcomes from the discussions in the national stakeholder meeting.

Our study, however, shares the common limitation of implementation research, which is the ability to generalise the obtained knowledge of AMS implementation to different healthcare settings. Our evidence in two provincial hospitals in Vietnam will be generalisable to the community hospitals with medium levels of resources and staff available for implementation but lack of staff training and capacity in carrying out AMS activities. Nevertheless, the nature of implementation research in recognising and tailoring activities to the local contexts is an important aspect in bridging the gap in our understanding of what is known to be effective and what works in the actual patient care settings.[18] Further implementation research in other hospital and healthcare settings is required to provide more data to fill the know-how gap in AMS programmes in LMICs.

In conclusion, AMS implementation requires an in-depth understanding of the local conditions and practices and the contextualisation of the intervention strategies in Vietnam and similar LMIC settings. We showed that using implementation research design is an effective approach in assessing the feasibility and evaluation of the AMS programme. It also helps local hospitals understand their needs and start prioritising AMS in the local agenda, paving the solid steps for a long-term institutionalisation and improvement of the programme. As evident in our process, it is important to develop the skills and capacity of the multidisciplinary AMS team and identify creative combinations of strategies that could be carried out with locally available resources for maximal and sustainable outcomes.

**Author affiliations**
[1]Ha Noi Unit, Oxford University Clinical Research Unit, Ha Noi, Vietnam
[2]National Hospital of Tropical Diseases, Hanoi, Vietnam
[3]Hospital for Tropical Diseases, Ho Chi Minh City, Vietnam
[4]Viet Tiep Hospital, Hai Phong, Vietnam
[5]Dong Thap Hospital, Cao Lanh-Dong Thap, Vietnam
[6]Duke Antimicrobial Stewardship Outreach Network, Duke Center for Antimicrobial Stewardship and Infection Prevention, Duke University, Durham, North Carolina, USA
[7]Centre for Tropical Medicine and Global Health, Nuffield Department of Clinical Medicine, University of Oxford, Oxford, UK

**Acknowledgements** We acknowledge the invaluable support and collaboration of the Medical Services Administration of the Vietnam Ministry of Health and the World Health Organisation Office in Vietnam during the planning of this study.

**Contributors** VTLH, HRvD and DA obtained funding and contributed to all aspects of the study design, with input from LMQ, VHV and CMD. VTLH and HRvD had overall responsibility for the study. TTTH, VHV, VTHDE, TTDN and HPT coordinated the running of the study with support from NTH, TD, PVBB, NTTO, PTTH, NTCT and TAQ. VTLH analysed the data with input from HRvD, DA, EDA and TK. VTLH was responsible for the drafting of the manuscript. All authors gave approval for the final version of the manuscript.

**Funding** Pfizer Independent Grants for Learning & Change (IGLC) provides all project funding and The Joint Commission provides administrative oversight for the programme; VTLH was also supported by the National Institute for Health Research (NIHR) (using the UK's Official Development Assistance (ODA) Funding) and Wellcome (Grant Reference Number: 216367/Z/19/Z) under the NIHR-Wellcome Partnership for Global Health Research. The views expressed are those of the authors and not necessarily those of Wellcome, the NIHR or the Department of Health and Social Care.

**Competing interests** None declared.

**Patient and public involvement** Patients and/or the public were not involved in the design, or conduct, or reporting or dissemination plans of this research.

**Patient consent for publication** Not required.

**Ethics approval** The study protocol was approved by the Oxford University Tropical Research Ethics Committee (OxTREC Reference 526-19), and the Ethics Committee of National Hospital of Tropical Diseases (08/HĐĐĐ-NĐT 31 May 2019). The conduct of this study conformed to the principles embodied in the Declaration of Helsinki.

**Provenance and peer review** Not commissioned; externally peer reviewed.

**Data availability statement** All data relevant to the study are included in the article or uploaded as supplementary information. Summary data from this study could be made available upon reasonable request.

**ORCID iDs**
Vu Thi Lan Huong http://orcid.org/0000-0002-9579-5576
Vu Hai Vinh http://orcid.org/0000-0001-6130-7864

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
