## [Reviewer comments · BMJ Open]

ARTICLE DETAILS

TITLE (PROVISIONAL)	Assessing feasibility of establishing antimicrobial stewardship programs in two provincial-level hospitals in Viet Nam; an implementation research study
AUTHORS	Vu, Huong; Ta, Thi Dieu Ngan; Huynh, Phuong Thao; Le, Minh Quang; Tran, Thi Thu Hanh; Nguyen, Thi Hien; Tran, Duc; Vu, Hai Vinh; Chau, Minh Duc; Vo, Thi Hoang Dung Em; Phan, Van Be Bay; Nguyen, Thi Thuy Oanh; Pham Thi Thuy, Hang; Nguyen, Cam Tu; Truong, Anh Quan; Kesteman, Thomas; Dodds Ashley, Elizabeth; Anderson, D; van Doorn, H. Rogier

VERSION 1 – REVIEW

REVIEWER	Charani, Esmita Imperial College London, Centre for Infection Prevention and Management
REVIEW RETURNED	27-Jun-2021

GENERAL COMMENTS	Thank you for this manuscript from a team with a strong track record in this field. My comments are in the order they appear in the manuscript and not in order of importance. It is important to generate such evidence from different settings in LMICs and the team have done an excellent job. Abstract - well written and clear, my only point is perhaps in the methods the steps take to contextualise the initiative should be made clearer so that the conclusion is better understood. Strengths and limitations Actually the first point in the strengths could be articulated in the abstract to address my point above. Having said that, the bullet points here should be written slightly differently e.g. Using a participatory and teamwork approach that considered hospitals as a complex adaptive system with unique characteristics.... Employment of intervention strategies that could... Included perspectives etcetc Introduction - well written and relevant to the work, no comments from me.
--

Methods: What were the specialties of the clinical wards included in each hospital? Since selection was based on willingness to participate, did you have any refusals and did you collect data on reason for refusing to participate? Since the selection of this was left at discretion of the directors in the hospital and the wards in control were not matched to the intervention arm there may be issues with being able to compare them.

Page 6 bottom of page - should be TRAINING AND MENTORING not trainings and mentoring. Also did the DASON programme have to be adapted for Viet Nam? You are describing the steps you took to prevent decline effect and you have selected the correct approach, it would just be helpful to state how and if the training was tailored to local need. How often was training provided to whom and how?

Page 7 second to last para: Data were COLLECTED from the existing...

Discussion

It is very encouraging to increasingly see the recognition of the role of pharmacists in AMS. There has also been examples from India on pharmacy driven AMS initiatives:

<https://www.mdpi.com/2079-6382/10/2/220>

Did the pharmacists in this study receive any specific training to carry out their roles in the AMS? Particularly the PAF? The section on PAF and pharmacists in the discussion reads a bit like results, it should be more an interpretation of the results and not repetition of the figures and data. The section on the pharmacy vs microbiology driven AMS may hold true for Viet Nam and indeed some other countries, but equally in many part of the world pharmacists remain excluded. ID and microbiology are central to leading AMS through their expertise and knowledge and need to work closely with pharmacists to drive AMS. The authors have discussed sustainability of such initiatives where there is no external funding - can they make any recommendations for how one can make such efforts sustainable. The association with OUCRU will also be a positive and hence I do wonder if we need to advocate for greater international collaboration and partnerships to facilitate shared expertise and knowledge for AMS..

Tables and figures

I just want to clarify were all the proposed interventions in Table 1 implemented? In which case the last column should be Interventions implemented. And if they were not can you bold or highlight the ones that were implemented?

Table 2 : What period is the data from? Jan -Dec 2019? Can this be clarified please and how is this in relation to the timeline of interventions implemented?

Figure 2 again - need to give the time period- assuming Jan -Dec it just needs to be added to table legend or title for clarity

REVIEWER

Bronkhorst, Elmien
Sefako Makgatho Health Sciences University

REVIEW RETURNED	28-Jul-2021
GENERAL COMMENTS	Very well written paper, on a topical and interesting topic. Few corrections made on manuscript. Inconsistencies in using numerical values and writing out numbers

VERSION 1 – AUTHOR RESPONSE

***Reviewer: 1

Dr. Esmita Charani, Imperial College London

Response: We thank the reviewer for this comment.

***Abstract - well written and clear, my only point is perhaps in the methods the steps take to contextualise the initiative should be made clearer so that the conclusion is better understood.

Response: We have reformatted the abstract following the journal's required style and include more specific details in the study design and interventions to illustrate the steps in contextualizing the interventions using locally available resources.

***Strengths and limitations

Actually the first point in the strengths could be articulated in the abstract to address my point above. Having said that, the bullet points here should be written slightly differently e.g. Using a participatory and teamwork approach that considered hospitals as a complex adaptive system with unique characteristics....

Employment of intervention strategies that could...

Included perspectives etc, etc

Response: We have mentioned the first point in the abstract, and revised the writing of the section following the recommendation from the editor.

***Introduction - well written and relevant to the work, no comments from me.

Response: We appreciate this comment from the reviewer.

***Methods: What were the specialties of the clinical wards included in each hospital? Since selection was based on willingness to participate, did you have any refusals and did you collect data on reason for refusing to participate? Since the selection of this was left at discretion of the directors in the hospital and the wards in control were not matched to the intervention arm there may be issues with being able to compare them.

Response: We have included a list of specialties in the Methods section under the Implementation approach subheading. There was no refusal for participation and the assignment of the clinical wards to the intervention and control group was discussed internally within each hospital board and AMS meeting. Since AMS implementation requires collaboration and willingness of the wards, we need to allow the hospitals the flexibility to be able to decide on the wards that are more likely to receive the interventions.

This is the nature of implementation research design where we need to factor in the real world situations in the local hospitals. Below is the added text in the manuscript in response to this comment:

“At both hospitals, the directors and the AMS teams decided to select the wards that were more likely to accept the AMS interventions based on their judgment and internal relationships. The allocation of the wards to each study group was as follows:

At Hospital 1:

- Intervention group included surgical intensive care unit, traumatology, respiratory/ musculoskeletal system, and infectious diseases ward;
- Control group included internal intensive care unit, surgical nephro-urology, pediatrics and general internal medicine ward;

At Hospital 2:

- Intervention group included surgical intensive care unit, general internal medicine, traumatology, and high-quality-serviced ward (provide treatment services with better quality based on patient’s request);
- Control group included internal intensive care unit, oncology, infectious diseases and surgical gastroenterology ward”

***Page 6 bottom of page - should be TRAINING AND MENTORING not trainings and mentoring. Also did the DASON programme have to be adapted for Viet Nam? You are describing the steps you took to prevent decline effect and you have selected the correct approach, it would just be helpful to state how and if the training was tailored to local need. How often was training provided to whom and how?

Response: We have now created a separate section on Training and mentoring to describe this component of the implementation process in more detail. We incorporated both international training by DASON onsite both in the US and Viet Nam to provide the opportunity for participants to learn about the international models and discuss the applicability of the models to the Vietnamese context. Training was provided to local staff throughout the implementation, with topics discussed between all partners involved and timing was decided to suit the timelines at each hospital and was also dependent on the occurrence of COVID-19 outbreaks. Below is the added text to provide more information on the training and mentoring in this section:

“The following training events were organized throughout the planning and implementation phase, with topics discussed and agreed upon by all partners involved and timing decided based on practical considerations, including the impact of COVID-19 on routine care in each location:

- A study tour to Duke University was organized for AMS staff of two study hospitals and local AMS experts from NHTD and HTD in July-August 2019. Participants attended lectures presented by DASON experts on AMS principles and practical guidance and experiences on specific AMS interventions. Participants also visited four DASON community hospitals to learn about the AMS models implemented at these sites. Participants also discussed about the feasibility and applicability of the interventions to the Vietnamese context.
- Visits by DASON experts and OUCRU staff at each study hospital in Viet Nam were organized in December 2019 to provide on-site training and technical support in assessing the needs and planning of activities.
- Training courses by local experts on antibiotic treatment, clinical pharmacy and clinical microbiology were organized in a class-room style for doctors, pharmacists and microbiologists. These courses were provided at two time points: one initial course in June 2020 to cover the broad concepts and one follow-up course in October 2020 focusing on the specific needs and knowledge gaps of each hospital.

- One-week training and shadowing for clinical pharmacists on prospective audit and feedback (PAF) at HTD was organized in December 2020.
- Training course on microbiological specimen sampling was organized at Hospital 1 for nurses in April 2021.
- Training and on-going technical support were provided by OUCRU for AMS staff on specific skills in planning and implementation of AMS activities.

Local meetings were organized in each hospital with the participation of AMS team members and participating intervention wards to assess the current gaps and needs and develop plans for implementation. Mid-term meetings and project monitoring meetings were also organized by OUCRU to review on the progress and upcoming activities with the AMS team at each hospital.”

***Page 7 second to last para: Data were COLLECTED from the existing...

Response: We have revised this accordingly.

***Discussion

It is very encouraging to increasingly see the recognition of the role of pharmacists in AMS. There has also been examples from India on pharmacy driven AMS initiatives: <https://www.mdpi.com/2079-6382/10/2/220>

Did the pharmacists in this study receive any specific training to carry out their roles in the AMS? Particularly the PAF?

Response: Our pharmacists received both class-room training on site and visits to Hospital for Tropical Diseases (HTD) to learn about the process and shadow HTD clinical pharmacists in performing PAF activity. This is now described in the training and mentoring section in the Methods section. The example from India that the reviewer referred to has now been added as a reference (number 28) to the text as follows:

“Sustained reductions have been achieved in antibiotic use by AMS teams across different settings including the example of non-specialist pharmacist-led program in South Africa²⁶ and clinician-led and clinical pharmacist-driven AMS program in India.²⁷”

***The section on PAF and pharmacists in the discussion reads a bit like results, it should be more an interpretation of the results and not repetition of the figures and data.

Response:

We have now updated the data and described these in the Results section under the section “Baseline analysis at two hospitals and implementation plan” as follows:

“By the end of the implementation period, 1890 antibiotic prescriptions were audited in Hospital 1 (time period: June 2020-May 2021; 1.5 fulltime clinical pharmacists) and 1628 in Hospital 2 (July 2020-July 2021; four fulltime clinical pharmacists). Recommendations were made in 82/1890 (4.3%) of the audited cases in Hospital 1 and 128/1628 (7.9%) in Hospital 2. The hospitals planned to continue PAF in the daily work of the clinical pharmacists.”

***The section on the pharmacy vs microbiology driven AMS may hold true for Viet Nam and indeed some other countries, but equally in many part of the world pharmacists remain excluded. ID and

microbiology are central to leading AMS through their expertise and knowledge and need to work closely with pharmacists to drive AMS. The authors have discussed sustainability of such initiatives where there is no external funding - can they make any recommendations for how one can make such efforts sustainable. The association with OUCRU will also be a positive and hence I do wonder if we need to advocate for greater international collaboration and partnerships to facilitate shared expertise and knowledge for AMS.

Response: We highly appreciate this comment and agree with the reviewer. However, in the Vietnamese context, the roles of both clinical pharmacy and microbiology are defined in the national AMS guideline and we would like to have a balanced discussion on both roles.

In terms of recommendations for sustainability, we have discussed this in our discussion section and point to the need to make use of local resources and integrate activities (particularly PAF) into routine work of hospital staff. Regarding the recommendations on more international collaboration and partnerships to facilitate shared expertise and knowledge on AMS, we have touched on this in the second paragraph of the discussion section. We added one more sentence (underlined) to this paragraph as follows to make the point clearer following the comment of the reviewer:

“Secondly, the partnership approach combining local and international partners in our study demonstrated an effective synergy of experience, expertise and resources to ensure the appropriate selection of strategies and methods in setting up AMS programs in LMICs. This approach has been advocated in a recent call for international collaborations in AMS implementation in LMICs, which can be achieved through the dissemination of high-quality research and educational resources recognizing the needs to contextualize to the local conditions.²⁶ Our study provided further evidence to support this call for greater international collaboration and partnerships to facilitate shared expertise and knowledge for AMS.”

***Tables and figures

I just want to clarify were all the proposed interventions in Table 1 implemented? In which case the last column should be Interventions implemented. And if they were not can you bold or highlight the ones that were implemented?

Response: Yes the interventions were implemented. We changed the name of column heading accordingly. There was one exception about the proposed activity of “Support from DASON on using web-based data visualization and benchmarking tool” which was noted as delayed due to discussions on data sending outside of Viet Nam and where the tool could be located.

***Table 2: What period is the data from? Jan -Dec 2019? Can this be clarified please and how is this in relation to the timeline of interventions implemented?

Figure 2 again - need to give the time period- assuming Jan -Dec it just needs to be added to table legend or title for clarity

Response: We have added time periods in both Table 2 and Figure 2. Timelines of interventions were also added to the Methods and Results section for clarity.

***Reviewer: 3

Dr. Elmien Bronkhorst, Sefako Makgatho Health Sciences University

Response: We thank the reviewer for their comments. We have addressed the inconsistencies in using numerical values and writing out numbers in the manuscript.